# Necroptosis in Solid Organ Transplantation: A Literature Overview

**DOI:** 10.3390/ijms23073677

**Published:** 2022-03-27

**Authors:** Beatrice Lukenaite, Erika Griciune, Bettina Leber, Kestutis Strupas, Philipp Stiegler, Peter Schemmer

**Affiliations:** 1General, Visceral and Transplant Surgery, Department of Surgery, Medical University of Graz, 8036 Graz, Austria; beatrice@juodas.com (B.L.); erikamilisauskaite@gmail.com (E.G.); bettina.leber@medunigraz.at (B.L.); peter.schemmer@medunigraz.at (P.S.); 2Faculty of Medicine, Vilnius University, 01513 Vilnius, Lithuania; kestutis.strupas@santa.lt

**Keywords:** ischemia-reperfusion injury, necroptosis, solid organ transplantation, sterile inflammation

## Abstract

Ischemia-reperfusion injury (IRI) is encountered in various stages during solid organ transplantation (SOT). IRI is known to be a multifactorial inflammatory condition involving hypoxia, metabolic stress, leukocyte extravasation, cellular death (including apoptosis, necrosis and necroptosis) and an activation of immune response. Although the cycle of sterile inflammation during IRI is consistent among different organs, the underlying mechanisms are poorly understood. Receptor-interacting protein kinase 3 (RIPK3) and mixed-lineage kinase domain-like pseudokinase (MLKL) are thought to be crucial in the implementation of necroptosis. Moreover, apart from “silent” apoptotic death, necrosis also causes sterile inflammation—necroinflammation, which is triggered by various damage-associated molecular patterns (DAMPs). Those DAMPs activate the innate immune system, causing local and systemic inflammatory responses, which can result in graft failure. In this overview we summarize knowledge on mechanisms of sterile inflammation processes during SOT with special focus on necroptosis and IRI and discuss protective strategies.

## 1. Introduction

Solid organ transplantation (SOT) has transformed the view to end-organ dysfunction. It not only changed the quality of life for the patients with terminal illness, but also offers a lifesaving treatment [1]. The beginning of transplantations era was in the 20th century, when the first successful kidney transplantation was performed on 23 December 1954 by Joseph Murray. He bypassed the barrier of rejection by using the patient’s identical twin as the donor of a human kidney transplant [2,3]. These experiences provided a foundation for discovering immunologic concepts for transplantation, including the knowledge of allograft rejection and the development of immunosuppressive therapy. New surgical techniques and immunosuppressive therapy allowed more efficient transplantations with fewer complications, decreased ischemic injury events and lowered the host immune response improving short and long-term graft survival [1]. However, long-term graft survival remains one of the most important issues.

In all SOT, organs are destined to be exposed to ischemia-reperfusion injury (IRI) in various stages, from organ retrieval and cold storage to implantation to the recipient. IRI can compromise the early and long-term success of the graft. IRI is multifactorial inflammatory condition involving hypoxia, metabolic stress, leucocyte extravasation, cellular death (including apoptosis, necrosis and necroptosis) and also an activation of immune response [4]. IRI induces significant injury to the graft including innate immune response in the recipient. All organs have specific mechanisms of IRI during transplantation, which will be the further objective in this review.

Current studies mainly focused on the role of the T cell adaptive immune response and lack of respective immunosuppression, which leads to innate immune response and grafts sterile inflammation (SI) and IRI. Many pathways are described which irreversibly damage the transplanted organ involving among others apoptosis and necroptosis. However, it is still not clear how necroptosis impacts adaptive immunity.

The aim of this overview is to overview the currently available literature on the known mechanisms of necroptosis and immune rejection in SOT.

## 2. Materials and Methods

The literature search was performed in the PubMed, Web of Science and EMBASE online databases. The following combinations for the search of the literature were used: “transplantation” OR “transplant” AND “necroptosis” OR “ischemia-reperfusion injury” AND “organ” OR “solid organ” OR “liver” OR “kidney” OR “lung” OR “heart” OR “pancreas”.

The accepted articles included clinical trials, meta-analysis, randomized control trials, reviews and systematic reviews. The time limit was chosen to be 10 years. The search was restricted to the English language. There were no other limitations. The most recent search was performed on 22 February 2022.

At least two researchers reviewed the abstracts. After relevant abstracts were identified, full-text articles were re-reviewed. Relevant articles from reference list of those selected studies were also included.

## 3. Literature Overviw

### 3.1. Apoptosis vs. Necrosis vs. Necroptosis vs. Ferroptosis

Cell death is a fundamental process in embryonic and neonatal development as well as in homeostasis in all organs in our body. Cell death is important in removing aged and damaged cells, preventing organ dysfunction and carcinogenesis. Even though there are many cellular death types, the major types dominating in SOT are necrosis, apoptosis and, more recently discovered, the regulated cell death type—necroptosis. These main cellular death types are summed up in Table 1. Another type of regulated necrosis—ferroptosis—will be also discussed briefly. There are many other types of cell death, however, they are not in the scope of this research.

#### 3.1.1. Apoptosis

One of the best-understood cellular death types is apoptosis, described as regulated cellular death. Apoptosis normally occurs during development and aging and as a homeostatic mechanism to maintain cell populations in tissues. Apoptosis also occurs as a defense mechanism such as in immune reactions or when cells are damaged by disease or noxious agents [5,6] and is characterized as a cascade of intracellular mechanisms leading to programed cell death. 

Apoptotic cell death is triggered either by the intrinsic (or mitochondrial) or extrinsic (death receptor) pathway. Both pathways eventually are initiated by initiator caspase-8 and caspase-9 activating executioner caspase-3. Caspase-3 is then cleaved at an aspartate residue to yield a p12 and a p17 subunit to form the active caspase-3 enzyme which is responsible for DNA fragmentation, degradation of cytoskeletal and nuclear proteins, cross-linking of proteins, formation of apoptotic bodies, expression of ligands for phagocytic cell receptors and uptake by phagocytic cells [6,7]. Caspases are widely expressed in an inactive form and once activated can activate other procaspases, initiating the protease cascade, which leads to inevitable cell death. To date, ten major caspases have been identified and broadly categorized into initiators (caspase-2, -8, -9, -10), effectors or executioners (caspase-3, -6, -7) and inflammatory caspases (caspase-1, -4, -5) [8]. Another biochemical feature of apoptosis is the expression of cell surface markers that result in the phagocytic recognition of apoptotic cells, permitting quick phagocytosis with minimal effect to the surrounding tissue [5].

The extrinsic pathway is initiated by the activation of the transmembrane “death domain”, which is a member of the tumor necrosis factor (TNF) receptor gene superfamily [9]. Binding of ligands and corresponding death receptors (FasL/FasR, TNF-α/TNFR1) results in the binding of the adapter protein Fas-associated death domain (FADD) or the TNF receptor-associated death domain (TRADD), correspondingly [10]. FADD then associates with procaspase-8. After the activation of caspase-8, the execution phase of apoptosis is triggered.

The intrinsic pathway is a non-receptor modulated pathway. Negative (e.g., lack of growth hormone) or positive (radiation, toxins, hypoxia, hyperthermia, viral infections, free radicals) stimuli cause changes in the inner mitochondrial membrane, resulting in pore formation, loss of mitochondrial transmembrane potential and release of pro-apoptotic proteins [11]. 

The extrinsic and intrinsic pathways both reach the execution phase, which is the same for both apoptosis pathways, with the activation of the execution caspases (Caspase-3, caspase-6, and caspase-7). They activate cytoplasmic endonuclease, which degrades nuclear material, and proteases, which degrade the nuclear and cytoskeletal proteins [5]. Phagocytic uptake of apoptotic cells is the last component of apoptosis.

#### 3.1.2. Necrosis

Necrosis, a quasi-unregulated type of cell death, occurs as a consequence of various extracellular events leading to cell damage and its unprogrammed death [12]. Unregulated cell death is accompanied by swelling of the cell, distention of cellular organelles, clumping and degradation of nuclear DNA, extensive plasma membrane endocytosis and autophagy [13]. Necrosis generally occurs as a consequence of sever changes in physiological conditions, such as hypoxia, ischemia, hypoglycemia, toxin exposure, exposure to reactive oxygen metabolites, extreme temperature changes and nutrient deprivation [9,10]. 

Mitochondrial permeability transition (MPT)-driven necrosis is one of the forms of regulated cell death initiated by specific disruptions of the intracellular microenvironment, such as severe oxidative stress and cytosolic Ca^2+^ overload, which manifests with a necrotic morphotype [14,15,16,17]. Even though MPT is a type of regulated necrosis, MPT and other types of regulated necrosis including ferroptosis and necroptosis seem to operate independently form each other at least in pathological settings [18]. These observations in literature suggest that MPT-driven regulated necrosis evolved more closely to MOMP-dependent apoptosis than necroptosis, reflecting their different reliance on mitochondria [19]. Mitochondria tightly regulate calcium ion concentration in the mitochondrial matrix. Cellular hypoxia during ischemia leads to increased intracellular Ca^2+^. This increase together with oxidative stress opens mitochondrial permeability transition pores (MPTP) in the inner mitochondrial membrane and allows for the efflux of protons and loss of the pH gradient [20]. Opening of MPTP leads to water influx, mitochondrial swelling and the rupture of mitochondrial membranes, followed by the release of sequestered cell death factors and ROS from mitochondria [21]. This eventually results in a loss of cell membrane integrity and the release of the cytoplasmic contents into the surrounding tissue causing local inflammation.

#### 3.1.3. Necroptosis

Another more recently discovered cell-death pathway is necroptosis. This type of cell death has several non-regulated cell death morphological features such as organelle swelling, plasma membrane rupture, cell lysis, and leakage of intracellular components, which later cause secondary inflammatory responses [22,23]. Additionally, necroptosis represents an inflammatory model of cell death, which is similar to non-regulated necrosis [24]. However, necroptosis is regulated by various proteins, precisely receptor-interacting protein kinases 1 and 3 (RIPK1 and RIPK3) as well as the downstream substrate pseudo kinase mixed-lineage kinase domain like (MLKL) [11,14]. 

The necroptosis pathway can be initiated by ligand-dependent stimulation of cell surface death receptors, such as Fas, tumor necrosis factor (TNF) receptor 1 (TNFR1), IFN receptors (IFNRs), toll-like receptors (TLRs), and intracellular RNA- or DNA- sensing molecules [25,26]. It is well known that the involvement of receptor-like Fas, TNFR, and TNF-related apoptosis-inducing ligand (TRAIL) can lead to cell death through the recruitment of caspase-8 which leads to initiation of the extrinsic apoptotic pathway [27]. A vast majority of studies show that inhibition of caspase-8 shifts the extrinsic apoptosis towards the necroptosis mode of the cell through activation of RIPK3 and MLKL [28,29,30]. While caspase-8 is blocked, initiation of necroptosis is mediated by immune ligands FasL, TNF, or Lipopolysaccharides (LPS), leading to activation of RIPK3 which further activates the MLKL by phosphorylation [31]. Phosphorylated MLKL is translocated into the plasma membrane and disturbs the integrity of the cell. MLKL acts in two ways: either it acts as platform in plasma membrane for recruitment of Na^+^ or Ca^2+^ ion channels or promotes the pore formation in the plasma membrane [32]. Two main conditions are needed for necroptosis onset: cells must express RIPK3 and have inhibition of caspase-8. A large number of in vitro studies have shown that inhibition of caspase-8 molecules resulted in activation of RIPK3 which plays a key role in necroptosis [28,33]. 

Based on the main driving factors, necroptosis can be divided into three categories: (i) extrinsic necroptosis stimulated by TNFα, (ii) intrinsic necroptosis stimulated by reactive oxygen species (ROS), and (iii) ischemia-mediated intrinsic necroptosis [32]. 

During apoptosis, secretion of cytokines is minimal or even absent. During necroptosis, it is a main event leading to inflammation. Nevertheless, the release of damage-associated molecular patterns (DAMPs) from cells is the primary way by which necroptosis stimulates the inflammatory response. Recent studies have shown that RIPK3 also activates the formation of the so-called necrosome which is formed because of cellular stress or microbial infection to activate caspase-1 and caspase-11. However, several studies have reported that MLKL is essential for RIPK3-dependent inflammation.

#### 3.1.4. Ferroptosis

Another type of newly discovered regulated necrosis is ferroptosis. It is triggered by intracellular phospholipid peroxidation that is morphologically, biologically and genetically different from other types of cell death and it is more immunogenic than apoptosis [34]. Ferroptosis is a unique form of non-apoptotic cell death that relies on iron and lipotoxicity and it is triggered by iron-catalyzed lipid peroxidation initiated by non-enzymatic (Fenton reactions) and enzymatic mechanisms (lipoxygenases [LOXs]) [35]. Morphologically, cells undergoing ferroptosis have a typical necrotic morphology: small dysmorphic mitochondria with reduced amounts of cristae, a condensed membrane, a ruptured outer membrane and no hallmarks of apoptosis [36]. Several years of research on ferroptosis revealed the crucial role of mitochondria through mitochondrial lipid, energy and iron metabolism and other regulatory processes. Ferroptosis is usually characterized as the excessive peroxidation of phospholipid membranes rich in polyunsaturated fatty acids through an iron-dependent mechanism that leads to cell death [37]. Moreover, ferroptosis is also characterized by the massive release of oxidized lipid mediators. Compared to apoptotic cells, which are immunologically silent, those undergoing necroptosis, and ferroptosis are more immunogenic, as they release inflammatory cytokines and DAMPs promoting inflammation [38].

### 3.2. Ischemia-Reperfusion Injury

In all SOT organs are destined to be exposed to IRI in various stages, from organ retrieval, cold storage to implantation to the recipient. IRI is a multifactorial inflammatory condition involving hypoxia, metabolic stress, leukocyte extravasation, cellular death (including apoptosis, necrosis and necroptosis) and also an activation of immune response [4]. The total tissue injury induced by IRI is divided into two parts: ischemia injury and reperfusion injury. 

Ischemic injury is caused by hypoxia and hyponutrition. The initial injury caused by IRI is due to hypoxia. Low oxygen levels deplete ATP in mitochondria forcing them to switch to anaerobic cellular metabolism. Decreased ATP impairs normal functioning of the Na^+^/K^+^ pumps, restricting Na^+^ inside the cell. It also impairs Ca^2+^ excretion. The intracellular Ca^++^ overload generates reactive oxygen species (ROS) and the activation of NADPH oxidase [4]. ROS and the Ca^2+^ calcium overload produced during the initial metabolic stress results in cell death associated with IRI. Moreover, IRI induces an accumulation of lactate, which decreases intracellular pH and causes denaturation in proteins. In addition, ROS also mediates lipid peroxidation and destruction of the cell membrane, while the Ca^2+^ increase inside mitochondria leads to membrane instability. 

At the reperfusion stage, when the blood supply is re-established to ischemic tissue, the generation of reactive oxygen species (ROS) is increased due to decreased levels of antioxidative agents in ischemic cells. ROS cause oxidative stress, promoting endothelial dysfunction, DNA damage and local inflammatory responses. Inflammatory cascades and oxidative stress may afterwards induce a cytokine storm, which causes damage to cellular structures resulting in cell death [39].

The cell response to the injury depends on the severity of total tissue injury [40]. Prolonged ischemia reperfusion injury can lead to apoptosis, autophagy, necrosis, and necroptosis. To avoid cell death and injury to the graft, optimizing organ condition prior to the transplantation is necessary. Therapeutic and non-therapeutic strategies to protect the organ from IRI will be analyzed and described afterwards.

### 3.3. Transplantation and Sterile Inflammation

Sterile-inflammation is a form of pathogen-free inflammation caused by various stimuli such as mechanical trauma, ischemia, stress or environmental conditions such as ultra-violet radiation. These damage-related stimuli induce the secretion of DAMPs and other cytokines such as high-mobility group box-1 (HMGB1) extracellularly, which are recognized by innate immune receptors such as TLRs, which in turn mediate sterile inflammatory responses. Intracellular cytokines, such as interleukin-1α (IL-1α), are also important mediators of the sterile inflammatory response and can be released in their biologically active forms from necrotic cells [41]. Despite the growing knowledge of stimuli to induce sterile inflammation, the mechanisms how these stimuli trigger innate immune response is still not fully understood. On cellular level, the immune responses to infection and inflammation induced by sterile stimuli are similar. They include the recruitment of neutrophils and macrophages, the production of inflammatory cytokines and chemokines, as well as the induction of T-cell-mediated adaptive immune responses [42]. 

During transplantation the graft injury due to sterile inflammation is primarily associated with innate immune response; however, there is growing evidence suggesting that the adaptive immune system is also involved. Further we will analyze the specifics of sterile inflammation and necroptosis in different SOT settings.

#### 3.3.1. Liver Transplantation

First attempts in liver transplantation were done in 1963 by Starzl [43]. However, patients died after 22 and 7.5 days, shortly after the operation. After recognizing early setbacks, which included tissue ischemia, immunosuppression and coagulopathy challenges regarding biliary reconstruction, patients were surviving more than a year after the transplantation [43]. Even with new redefined surgical techniques and immunosuppression, long-term graft survival remains one of the most important issues. During liver transplantation, IRI is inevitable, and not only damages the graft, but is also associated with distal organ damage resulting from activation of the immune system [44]. 

The ischemic phase of IRI in liver transplantation contributes to 10% of early graft failure [45] and is also partly responsible for non-anastomotic biliary strictures [46,47]. At the cellular level, ischemia and reduced blood flow results in hypoxia in liver sinusoidal endothelial cells (LSEC), which subsequently triggers anaerobic metabolism in hepatocytes and causes ATP depletion in mitochondria [48]. This results in ADP, lactic acid accumulation and decreased pH levels in mitochondria. Due to depleted ATP levels, the function of ATP-dependent sodium potassium (Na^+^-K^+^) pump channel changes resulting in intracellular Na^+^ accumulation. Hyperosmolarity, caused by the accumulation of hydrogen (H^+^), Na^+^ and Ca^2+^ ions, lead to edema and swelling of hepatocytes, Kupffer cells and sinusoidal endothelial cells [49,50].

During the reperfusion phase, as the blood flow to the liver is restored, the cell injury is elevated due to a burst of ROS from mitochondria. Further Kupffer cell activation at this stage increases the release of ROS and proinflammatory cytokines, including tumour necrosis factor alpha (TNFα), interlukin-1 (IL-1), interferon-γ (IFN-γ) and IL-12 [51]. In normal situation the small amount of oxygen is reduced to ROS and neutralized in mitochondria by the antioxidant superoxide dismutase (MnSOD). However, during ischemia, due to excessive ROS production, MnSOD is not able to neutralize ROS, which results in oxidative stress causing endothelial dysfunction and DNA damage. Cell death results in the release of all cell organelles into the surrounding area, causing necrotic-type cell death and local damage inflammatory responses [52]. 

Apoptosis and necrosis are the two most important types of cell death. Also, necroinflammation is proven to determine the fate of liver graft and the recipient. Necroptosis is the main trigger of necroinflammation, though its role in the IRI is still under investigation. Following hepatic IRI, the release of proinflammatory cytokines and DAMPs cause sterile inflammation in the transplanted liver and activates innate immune response which contributes to graft rejection [53].

#### 3.3.2. Kidney Transplantation

The first successful human kidney transplant was performed between identical twins at the Peter Bent Brigham Hospital in Boston in December of 1954. Nowadays kidney transplantation is routinely performed as the possible choice of treatment to improve quality of life and survival for patients with end-stage renal disease.

Among the numerous variables that can affect the outcome of the transplanted kidney, one of the most common risk factors is IRI. Although IRI can also occur in the kidney transplanted from live donors, it is more frequent and severe in organs originated from deceased donors. IRI results in a rapid decline in kidney function and increased patient morbidity and mortality.

IRI is harmful to kidney transplants, resulting in delayed graft function (DGF) [54], acute and chronic rejection [55], and fibrosis. DGF is influenced by both cold and warm ischemia [56]. Endothelial cells and tubular epithelial cells are the first cells to suffer from reduced oxygen supply. Anaerobic glycolysis and ATP depletion lead to hyperosmolarity and ROS formation. The restorations of the blood flow in the reperfusion phase results in excessive ROS production ad inflammation. White blood cells, carried by the returning blood, release a wave of cytokines, chemoattractants, pro-coagulant factors and free radicals in response to tissue damage [56]. The severity of acute kidney injury (AKI) can influence the graft outcome. After a mild AKI, the kidney can repair itself. However, when the injury is more severe, the repair process can lead to fibrosis, which can progress to chronic kidney disease. 

Moreover, endothelial cells are the main target of both DGF and rejection. Endothelium reacts to any type of injury by remodeling the vascular wall. This process involved cell death and growth, cell migration and degradation or productions of cellular matrix. These changes result in intimal accumulation of smooth muscle-like cells and associated extracellular matrix, medial smooth muscle cell degeneration, adventitial fibrosis and compromised luminal flow [56]. IRI is an inevitable event after deceased donor transplant and can heavily influence both early and the function of a kidney allograft.

#### 3.3.3. Lung and Heart Transplantation

In lung transplantation, organ ischemia and following reperfusion is inevitable and usually leads to acute, sterile inflammation after transplant, which is also called ischemia-reperfusion (IR) injury. It is a major clinical issue, because it leads to primary graft dysfunction (PGD), short- and long-term morbidity and mortality [57,58]. 

Although reperfusion of the graft is necessary to avoid irreversible ischemic damage, it paradoxically also promotes further damage and dysfunction. IRI is a complex inflammatory response which involves endothelial and epithelial injury/dysfunction, release of cytokines and DAMPs, and innate immune responses including activation of alveolar macrophages, invariant natural killer T (iNKT) cells and neutrophils [59]. Most of the responses are duo to rapid generation of ROS. On physiological level ischemia, with or without anoxia, hypoxia in the arterioles and capillaries induce macrophages, endothelial cells, and other immune cells to generate ROS [60]. Activation of NADPH and proinflammatory cytokines causes the upregulation of cell-surface adhesion molecules on the endothelial side of the lung [57]. These actions cause physiological changes to the microvasculature and lead to pulmonary vascular resistance (PVR), pulmonary edema and oxygen exchange abnormalities. Infiltration of circulating host neutrophils into the graft is a key aspect of IR injury largely driven by potent chemokines, (e.g., IL-8 and CXCL2) produced by donor lung cells such as epithelium, endothelium or macrophages [61]. Neutrophil recruitment, pulmonary macrophages, NK cells and complement system activation are also linked to the IRI and later with PDG leading to chronic lung allograft dysfunction. It is important to note that innate immune responses can also increase adaptive immune responses inflammation after transplantation [62,63]. 

Analogous to lung transplantation, ischemia to the myocardium leads to injury and cell death from a combination of apoptosis and other regulated death pathways, including ferroptosis, necroptosis, and pyroptosis [64]. With the release of DAMPs, HMGB1 causes the activation of macrophages and the adaptive immune system, inducing production of cytokines and sterile inflammation [65,66].

Lung and heart IRI is a complex condition involving oxidative stress and subsequent responses by all cells within the lung, leading to breakdown of the endothelial and epithelial barriers resulting in life-threatening edema and defective gas exchange in the lung allograft [59].

### 3.4. Treatment Possibilities

Therapeutic strategies to limit IRI in transplantation generally fall into four categories: (i) reducing effects of ROS, (ii) modulation of the cytokine response, (iii) blocking activation of the immune system and (iv) improving organ preservation.

Some therapeutic strategies regarding modulation of the cytokine response and sterile inflammation suggest:Given the crucial role for IL-1 in sterile inflammatory responses, blockade of IL-1R has been tested as a therapeutic target for sterile inflammatory disorders in humans [41].As TLRs also mediate sterile inflammation, they could be another potential therapeutic target. However, studies in TLR2- and TLR4-deficient mice reported increased mortality in response to hypoxia-induced lung injury [67].Previous studies based on necrostatin-1 (Nec-1) demonstrated that inhibition of RIPK1 prevented cell death including necroptosis and apoptosis in an animal model of degenerative diseases [68]. Also, the administration of Nec-1 to both donor and recipient improved graft function in Lewi rats after lung transplantation through the reduction of necroptosis [69]. Nec-1 is the first extensively used compound identified as an inhibitor of necroptosis acting on the kinase activity of RIPK1 [22,70]. Moreover, results of study done on rats show, that the potency of Nec-1 to specifically interrupt necroptotic signaling provide a new strategy to prevent and treat ischemic kidney injury [71].Caspase inhibitors, such as the broad-spectrum caspase inhibitor zVAD-fmk or small-spectrum caspase inhibitors for caspase-8, are also under investigation. These molecules might be able to decrease IRI by blocking the apoptotic cascade during ischemia and so improving the graft function [57].Complement inhibition includes the inhibition of signaling to the leukocytes involved in the inflammatory response, as well as limiting the ability of the host to recognize the graft [57]. However, further research is needed to see if this is the case.Reducing IRI could also be done by converting necroptosis to apoptosis. Research done on rats shows that δV1-1, a protein kinase C (PKCδ) peptide inhibitor, reduced translocation of PKCδ and p53 to the mitochondria after 18 h of cold ischemia time. Administration of δV1-1 effectively reduced lung transplantation and IR-induced pulmonary injury in rats [72,73].

More therapeutic agents such as anti-TNFα biologic agents, prostaglandins E-1 and I2, antioxidant strategies using free radical scavengers or inhibitors of oxidant-producing enzymes, growth factors are being tested in the field of reducing IRI in SOT. However, more studies need to be done as the findings at this point are not promising in terms of good therapeutic abilities.

During transplantation, donor organ preservation is a critical step to maintain the quality of the organ as well as to improve post-transplant outcomes. Conventional static cold storage (SCS) is considered to be the primary method [74]. However, this method does not provide sufficient protection against IRI. Alternative strategies, such as machine perfusion (MP), have been introduced to reduce the harmful effects of SCS and to assess the organ quality [75,76,77]. Machine perfusion methods not only mimic the physiological process by establishing controlled continuous flow of nutrients and antioxidants, but also promote the flushing of inflammatory cytokines and toxins from the graft [78]. Nowadays MP is considered to be superior to SCS in terms of reducing DGF [79].

Moreover, the pharmacologic preconditioning of the brain-dead donor has been shown to ameliorate the allo-immune response [80]. Preconditioning with calcineurin inhibitors (CNIs) has been shown to have protective effects in a model of renal transplantation in rats compared to vehicle-treated animals [81]. Also postconditioning with dopamine derivate n-octanoyl-dopamine (NOD) through downregulation of the pro-apoptotic factor caspase-3, pro-inflammatory cytokines, and NF-kappaB may protect the heart against the myocardial injuries associated with brain death and ischemia/reperfusion [80].

## 4. Conclusions

Many studies are expanding our knowledge and view on necroptosis and sterile inflammation in SOT. However, necroptosis remains a rather unexplored pathway in organ injury. While necroptosis originates from the body’s natural protection against infection, programed cell death has unquestionable adverse effects on organ graft function and provokes alloimmunity. As we know, RIPK3 and MLKL play fundamental roles in necroptosis in IRI, however further investigation of the potential crosstalk between various cell-survival and cell-death signaling pathways during sterile inflammation should be investigated. 

While most studies on IRI focus on the transplantation and reperfusion state per se, it should be clear that organ retrieval is a major trigger for inflammatory changes like IRI in the graft and thus plays a pivotal role for the later fate of the graft [82,83,84].

Until recently, necroptotic cell death has not been a focus in transplant studies. However, in our opinion, the role of necroptosis is significant in SOT and IRI. Even though this type of cell death has complex and overlapping pathways of action, necroptosis does not have a unique marker beyond the lack of caspase activation. As we presented in the review, necroptosis has an unquestionable significance in IRI and adverse effects on organ graft function and in provoking alloimmunity. Considering its importance in successful transplantation, necroptosis and, most importantly, treatment possibilities of necroptosis should be the main scope in future research.

In summary, necroptosis is a promising therapeutic target during graft preservation and should be further explored not only for our deeper understanding but also for the translation of the knowledge in medical practice.

## Figures and Tables

**Table 1 ijms-23-03677-t001:** Differences between apoptosis, necrosis and necroptosis.

	Apoptosis	Necrosis	Necroptosis
**Type of cell death**	Controlled	Uncontrolled	Controlled
**Triggers**	Development, self-renewal, aging, trauma, stress	Trauma, stress, infection	Trauma, stress, infection
**Morphology**	Extensive membrane blebbing, condensation and fragmentation of the nucleus	Extensive organelle and cell swelling, loss of integrity of the cell membrane, release of the intracellular contents	Cytoplasmic swelling, rupture of the plasma membrane and spilling of the intracellular contents
**Signaling pathway**	Intrinsic and extrinsic	Unspecific	Specific, e.g., TNRF1 pathway
**Executioner**	Caspase (caspase 3, 6, 7, 8, 9)	-	RIP kinase (RIPK1, RIPK3)
**Complex formed**	Apoptosome	-	Necroptosome
**Inflammation response**	Anti- or pro-inflammatory	Pro-inflammatory	Pro-inflammatory
**DAMP release**	Yes	Yes	Yes
**Inhibitor**	Z-VAD fmk	-	Necrostatin-1

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
