# Peer review of "Necroptosis in Solid Organ Transplantation: A Literature Overview"

_ijms, 2022, doi:10.3390/ijms23073677_

Round 1

Reviewer 1 Report

Comments on IJMS

February 19, 2022

Necroptosis in solid organ transplantation: a literature review

Necroptosis is a newly discovered type of programmed cell death. It has been noted from solid organ transplantation as a new potential mechanism. However, the present literature review does not address this interesting and potentially important research area. The authors first provided some basic knowledge on apoptosis, necrosis, necroptosis and ferroptosis. They introduced ischemia-reperfusion injury in solid organ transplantation. They then provided basic description on liver, kidney, heart and lung transplantation and give suggestions on how to manage ischemia-reperfusion injury in organ transplantation. However, they did NOT give any information about “necroptosis in solid organ transplantation”.

Major comments:

  1. Data collection by 5 years: Why only 5 years? There are important research articles published before this time period. Moreover, from the text and reference list, I cannot find any paper that described necroptosis in solid organ transplantation.
  2. I did literature search, and I find some research paper that described necroptosis in ischemia reperfusion injury and solid organ transplantation. For example:

Linkermann A, Bräsen JH, Himmerkus N, et al. Rip1 (receptor-interacting protein kinase 1) mediates necroptosis and contributes to renal ischemia/reperfusion injury. Kidney Int. Apr 2012;81(8):751-61. doi:10.1038/ki.2011.450

Linkermann A, Green DR. Necroptosis. N Engl J Med. Jan 2014;370(5):455-65. doi:10.1056/NEJMra1310050

Kim H, Zhao J, Zhang Q, et al. δV1-1 Reduces Pulmonary Ischemia Reperfusion-Induced Lung Injury by Inhibiting Necrosis and Mitochondrial Localization of PKCδ and p53. Am J Transplant. Jan 2016;16(1):83-98. doi:10.1111/ajt.13445

Kim H, Zamel R, Bai XH, et al. Ischemia-reperfusion induces death receptor-independent necroptosis via calpain-STAT3 activation in a lung transplant setting. Am J Physiol Lung Cell Mol Physiol. Oct 2018;315(4):L595-L608. doi:10.1152/ajplung.00069.2018

Kanou T, Ohsumi A, Kim H, et al. Inhibition of regulated necrosis attenuates receptor-interacting protein kinase 1-mediated ischemia-reperfusion injury after lung transplantation. J Heart Lung Transplant. Oct 2018;37(10):1261-1270. doi:10.1016/j.healun.2018.04.005

  1. All three figures are adopted from other review articles. They look almost the same as those from their original sources. They are not accurately reflect the current knowledge on cell death described in this paper. They are unnessary and should be deleted.

  1. The discussion on IR injury and major organ transplantation are common knowledge and there is nothing new. There are no mentions on necroptosis in these solid organ transplanation.

Minor comments:

There are many grammatic errors, for example:

  1. Page 4. The section of MPT-initiated regulated necrosis should be discussed together with necroptosis, not with necrosis.
  2. Page 4, line 154. TNF should be TNFR.
  3. Page 4, line 159. Fas should be Fas ligand (FasL).
  4. Page 5, line 165. Please edit this sentence. You mentioned “two main conditions are needed”, but then you said “either A or B” will be OK.
  5. Page 5, last paragraph. There are several errors. First “RIPK3 stimulates the inflammatory responses” should be “necroptosis stimulates the inflammatory responses”. Second, RIPK3 activates the formation of necrosome, not inflammasome. The inflmmasome mediates pyroposis, not necroptosis.

Author Response

Thank you for your valuable input regarding our manuscript.

We carefully considered your comments and implemented changes accordingly.

  • We expanded the time span of data collection to 10 years as you suggested.
  • We appended citations according to your suggestion.
  • Minor comments and grammatical errors were also corrected according to your suggestions.
  • Figures were deleted.

  • As the topic is very broad, we believe that the main information regarding necroptosis is concentrated in this literature review. Going into more detail, as you suggested, would of course be very interesting, but might elongate the manuscript making it hard to read as a whole. We believe, that this work is able to give the interested reader a good general overview and starting point for in-depth research regarding the topic.

  • Even though MPT is a type of regulated necrosis, MPT and other types of regulated necrosis including ferroptosis and necroptosis seem to operate independently form each other at least in pathological settings [1]. These observations in literature suggest, that MPT-driven regulated necrosis evolved more closely to MOMP-dependent apoptosis than necroptosis, reflecting their different reliance on mitochondria [2].

[1] Linkermann A, Bräsen JH, Darding M, Jin MK, Sanz AB, Heller J-O, et al. Two independent pathways of regulated necrosis mediate ischemia–reperfusion injury. Proc Natl Acad Sci U S A 2013;110:12024–9. https://doi.org/10.1073/pnas.1305538110.

[2] Galluzzi L, Kepp O, Kroemer G. Mitochondrial regulation of cell death: a phylogenetically conserved control. Microb Cell n.d.;3:101–8. https://doi.org/10.15698/mic2016.03.483.

Reviewer 2 Report

In this review the authors summarize knowledge on mechanisms of inflammation processes during SOT with a special focus on the necroptosis and IRI and discuss protective strategies. It’s very interesting work and authors are to be congratulated on presenting results of their bibliographic research, but I preferred that the authors describe others new ways of cellular death liked to inflammation such pyroptosis and describe also the implicated mechanisms and the therapeutic strategies to reduce I/R injury and associated cell death in solid organ transplantation as well as marginal donor (Brain-Dead Donors and steatotic livers…)

Author Response

Thank you for your kind words regarding our work.   As this literature review was based on necroptosis and its emergence from regulated and unregulated cell death, we didn’t get into such detail regarding other cell death types, as they were not the scope of our research.   The topic is broad and could be analyzed in a deeper level, however without delving into the details of every possible biochemical action, therapeutic strategies would be too hard to understand while reading such an extent review.   We carefully considered your comments and added a shot paragraph regarding treatment possibilities for brain-dead donors. 

Round 2

Reviewer 1 Report

I appreciate that the authors take my comments and revised the manuscript. However, as I mentioned in my previous review on this manuscript, the major weakness of this paper is that it does not provide enough information about “necroptosis in solid organ transplantation”.

They only added a few new references I provided. To my knowledge, there are many publications on this subject. If the authors really want to publish this paper, they need to collect these publications, analyze them, and present them with their critical comments. Without these, this paper is just a repeat of common knowledge.

Author Response

Thank you for your valuable comments. According to your suggestions, we corrected the title and the design of the manuscript. As this field is so extensive, we cannot guarantee that it would cover all topics of interest and provide a comprehensive review including all information available and therefore we decided to change the title to “literature overview” in order to avoid confusions.

However, in our opinion, this article gives an excellent overview of the necroptosis topic and provides a lot of reviewed articles that might be of interest for the readership. As there is a vast amount of possible literature, analyzing and adding all of them would create a manuscript difficult to follow and therefore we hope to be able to give an overview including most important and recent papers in the field.

In our opinion, the common knowledge and newly discovered information presented in this overview, gives a better understanding to the readers about the topic itself and its significance in the field.